🔓 | Open Peer Review | Virology | Research Article

# Blood transcriptomic profiling reveals gene expression alterations in patients with SFTS-associated encephalitis

DaiQing Wu,[1] AoFan Wang,[1] Junjie Shi,[2] Ying Zhang,[3] Yu Geng,[1] Huifang Liu,[4] Yuanyuan Wu,[5] Wenwen Kong,[5] Yijia Zhu,[1] Yuxin Chen[1,6]

**ABSTRACT** Severe fever with thrombocytopenia syndrome (SFTS), a life-threatening tick-borne zoonosis caused by severe fever with thrombocytopenia syndrome virus (SFTSV), frequently leads to fatal encephalitis characterized by consciousness disorders and seizures. The molecular mechanisms governing SFTSV neuroinvasion and host-driven neural injury remain largely elusive. To explore the mechanisms of SFTS-induced brain damage, we analyzed clinical laboratory parameters and conducted transcriptomic analyses of peripheral blood mononuclear cells from five SFTS patients with encephalitis and five non-encephalitis patients admitted to Nanjing Drum Tower Hospital during the same period. Our findings indicate that central nervous system manifestations in SFTSV infection are associated with altered expression of immune-related genes. Specifically, we identified six differentially expressed immune genes—MET, KIT, IL1R2, MAFF, CD69, and CEBPD—between the encephalitis and non-encephalitis groups. This study provides novel insights into the pathogenesis of SFTS-associated encephalitis, and further investigation into the host immune response post-SFTSV infection may aid in mitigating disease progression and improving clinical outcomes.

**IMPORTANCE** Severe fever with thrombocytopenia syndrome (SFTS) is a life-threatening disease that can lead to encephalitis—a serious brain inflammation with high mortality. However, the causes of this brain damage remain largely unknown. In this study, we used advanced gene sequencing techniques to analyze blood samples from SFTS patients with and without encephalitis. Our results revealed key changes in immune-related genes, uncovering possible biological pathways involved in brain injury caused by the virus. These findings shed new light on how the immune system may contribute to neurological complications in SFTS and highlight specific genes that could serve as future targets for diagnosis or treatment. This research enhances our understanding of SFTS-related encephalitis and provides a valuable foundation for developing therapies to improve patient outcomes.

**KEYWORDS** severe fever with thrombocytopenia syndrome, encephalitis, peripheral blood mononuclear cells (PBMC), transcriptomic analysis

Severe fever with thrombocytopenia syndrome (SFTS) is an emerging infectious disease characterized by its unique etiology, epidemiological patterns, and diverse clinical manifestations. The disease is caused by a novel bunyavirus, designated as severe fever with thrombocytopenia syndrome virus (SFTSV), which was initially identified in China in 2009 and subsequently has been reported in several countries in Southeast Asia (1–6). SFTS is primarily transmitted through tick bites, with the highest incidence occurring in rural areas during warmer seasons (7, 8). Clinically, SFTS presents with a sudden onset of fever, thrombocytopenia, leukopenia, and gastrointestinal symptoms, with severe cases potentially progressing to multi-organ dysfunction and fatalities (9–

Address correspondence to Yuxin Chen, yuxin.chen@nju.edu.cn.

DaiQing Wu, AoFan Wang, Junjie Shi, and Ying Zhang contributed equally to this article. The author order was determined based on their contribution to the article.

The authors declare no conflict of interest.

11). The mortality rate of SFTS is notably high, varying between 5% and 30% depending on geographical locations (11). In addition to its systemic effects, SFTS is also associated with the central nervous system (CNS), manifesting as encephalitis or encephalopathy in a significant proportion of patients (12–14). Neurological complications are particularly severe in this disease, with studies suggesting that a high proportion up to 44.7% of patients with SFTS-associated encephalitis (SAE) may not survive (15).

Despite increasing research on SFTS, studies specifically addressing SFTS-associated encephalitis remain limited. Previous investigations have primarily focused on the clinical features and epidemiology of SFTS, with less attention paid to the mechanisms underlying CNS involvement (9, 15, 16). The diagnosis of SFTS-associated encephalitis remains challenging due to the lack of definitive clinical markers and the infrequent collection of cerebrospinal fluid (CSF) samples. Moreover, the pathogenesis of SFTSV in the CNS remains poorly understood, with evidence suggesting both direct viral invasion and immune-mediated mechanisms could be involved (17). Although some studies have observed elevated cytokine levels in the CSF of patients with SFTS-associated encephalitis, a comprehensive understanding of the host immune response is still lacking (12, 18). This knowledge gap underscores the need for further investigations into the molecular and immunological mechanisms involved in SFTSV-associated CNS complications (19).

Viral infections often induce widespread changes in the host cell transcriptome, leading to metabolic dysregulation and immune dysfunction, which ultimately create a microenvironment conducive to viral replication (20, 21). Understanding the pathogenesis of SFTSV, particularly its impact on the CNS, is essential for developing effective therapeutic strategies and prognostic biomarkers. Peripheral blood mononuclear cells (PBMCs) are readily accessible sources of immune cells that can reflect the host immune response to viral infections (22, 23). Transcriptomic analysis of PBMCs provides valuable insights into the molecular changes that occur during SFTSV infection and the related pathogenesis of SAE. This approach is generally considered representative of the host immune response and could reveal key pathways and genes that are dysregulated during SFTSV infection. Importantly, there is currently a paucity of transcriptomic studies focusing on SFTS-associated encephalitis.

In this study, we characterized the transcriptome profile of PBMCs in patients with SFTS-associated encephalitis and non-encephalitis using high-throughput sequencing to identify alterations that occur during disease progression. Our objective is to explore the association between these changes and the CNS manifestations of SFTS. In our study, several signature immune biomarkers were identified that could improve the management of SFTS, particularly in patients with neurological complications.

## MATERIALS AND METHODS

### Enrollment of SFTS patients

This retrospective study included five patients diagnosed with SAE and five SFTS patients without neurological complications as controls. The sample size was determined based on clinical feasibility and the rarity of SAE cases, rather than formal statistical power calculations. All patients were diagnosed at Nanjing Drum Tower Hospital (Affiliated Hospital of Nanjing University Medical School) between May 2021 and August 2021 based on clinical presentation, epidemiological history, and laboratory confirmation. Laboratory diagnosis of SFTS was established through reverse transcriptase real-time PCR for SFTSV RNA detection in serum samples and serological testing using enzyme-linked immunosorbent assay or indirect immunofluorescence assay to detect SFTSV-specific IgM and IgG antibodies. The inclusion criteria for SAE were based on the presence of altered mental status, including symptoms such as headache, irritability, somnolence, and confusion, lasting for at least 24 hours, with other potential etiologies excluded. All patients were screened to exclude a history of cardiovascular diseases or malignancies. Demographic data and laboratory indices for all patients are systematically summarized in Table 1.

**TABLE 1** The demographic and clinical characteristics of the enrolled patients

| Characteristic | Encephalopathy ($n = 5$) | Non-encephalopathy ($n = 5$) | P-value[a] |
|---|---|---|---|
| Demographic characteristics | | | |
| Age (years) | 58 (57, 68) | 75 (72, 79) | 0.151 |
| Gender (male/female) | 2/3 | 2/3 | 0.738 |
| Underlying diseases, $n$ (%) | | | |
| Hypertension | 2 (40) | 3 (60) | 0.500 |
| Diabetes | 2 (40) | 3 (60) | 0.500 |
| Coronary heart disease | 0 (0) | 0 (0) | – |
| Chronic pulmonary diseases | 1 (20) | 1 (20) | 0.778 |
| Cerebrovascular disease | 0 (0) | 0 (0) | – |
| Malignancy | 0 (0) | 0 (0) | – |
| Clinical manifestation, $n$ (%) | | | |
| Fever | 5 (100) | 5 (100) | – |
| Maximum body temperature (°C) | 39 (39, 39.5) | 39 (39, 39) | 0.548 |
| Fatigue | 5 (100) | 5 (100) | – |
| Dizziness | 4 (80) | 1 (20) | 0.103 |
| Headache | 4 (80) | 2 (40) | 0.262 |
| Vomiting/diarrhea | 3 (60) | 3 (60) | 0.738 |
| Myalgia | 2 (40) | 1 (20) | 0.500 |
| Laboratory findings at sequencing submission | | | |
| DBV DNA $\log_{10}$(copies/mL) | 7 (5, 8) | 5 (5, 5) | 0.222 |
| Leukocytes ($\times 10^9$/L; normal range, 3.5–9.5) | 2.1 (1.7, 2.8) | 2.5 (2.4, 4.6) | 0.310 |
| Neutrophils (%; normal range, 40–75) | 65.7 (64.2, 69.1) | 80.6 (76.1, 85.6) | 0.008 |
| Lymphocytes (%; normal range, 20–50) | 20.5 (16.6, 28.1) | 10.3 (9.6, 19.1) | 0.095 |
| Monocytes (%; normal range, 3–10) | 12.9 (4.7, 13.9) | 4.1 (3.6, 5.5) | 0.222 |
| Erythrocytes ($\times 10^{12}$/L; normal range, 4.3–5.8) | 4.51 (4.41, 4.91) | 4.13 (3.47, 4.27) | 0.095 |
| Hemoglobin (g/L; normal range, 130–175) | 134 (132, 141) | 125 (114, 138) | 0.690 |
| Platelets ($\times 10^9$/L; normal range, 125–350) | 35 (25, 74) | 50 (46, 55) | 1.000 |

[a]"–" indicates that the two groups had identical numerical values for the variable under analysis.

## Blood sample collection and isolation of PBMCs

Peripheral blood samples from SFTS or SAE patients during the multiple organ dysfunction syndrome (MODS) stage were collected in either anticoagulant or clotting tubes and stored at 4°C. The blood was subsequently diluted, layered, and centrifuged using lymphocyte separation medium and phosphate-buffered saline to isolate mononuclear cells. After purification by low-speed centrifugation, the cell count was determined using a hemocytometer to obtain the required PBMCs. Isolated PBMCs were immediately cryopreserved at −80°C for subsequent RNA extraction and sequencing.

## RNA extraction, transcriptome library construction, and next-generation sequencing

Total RNA was extracted from PBMCs using QIAzol Lysis Reagent (QIAGEN, Hilden, Germany) and assessed for integrity and quantity using the Agilent 2100 Bioanalyzer system. Samples with an RNA integrity number greater than 7.5 were retained to ensure the reliability of subsequent sequencing results. Following the Illumina library construction protocol, mRNA was enriched using Oligo (DT) magnetic beads that bind the polyA tail. The mRNA was then randomly fragmented in the fragmentation buffer and used as a template for library construction. Oligonucleotide primers and M-MuLV reverse transcriptase were employed to synthesize the first and second strands of cDNA. cDNA quantification was performed using the Qubit 2.0 Fluorometer, and the library was diluted to a concentration of 1.5 ng/µL. The insert size of the library was assessed using the Agilent 2100 Bioanalyzer. Qualified libraries were pooled and sequenced on the Illumina NovaSeq 6000 platform, generating 150 bp paired-end reads. Basecaller

software was then used to convert optical signals into sequencing peaks to obtain the sequences of the target fragments.

## Bioinformatics analysis of the RNA-seq data

Raw sequencing data were processed for quality control and adapter trimming using FASTQ software to remove sequencing adapters and low-quality reads. The paired-end clean reads were then aligned to the reference genome using the HISAT2 aligner. Gene expression quantification was performed using the FeatureCounts tool. Differential gene expression analysis was conducted with the DESeq2 R package, and genes were considered significantly differentially expressed if they met the criteria of |log2 Fold Change| > 1 and a *P*-value < 0.05. Volcano plots and heatmaps of the most differentially expressed genes (DEGs) across comparison groups were generated using R with the EnhancedVolcano and heatmap packages. To further explore the biological significance of the differentially expressed genes, we performed gene ontology (GO) enrichment analysis and Reactome Pathway enrichment analysis. Functional annotation and pathway analysis were conducted using the online DAVID Functional Annotation Tools (https://davidbioinformatics.nih.gov/). A bubble plot was created using https://www.bioinformatics.com.cn. Gene set enrichment analysis (GSEA) was carried out using the fgsea R package. The CIBERSORT tool, based on deconvolution algorithms, was employed to estimate the composition and abundance of immune cells within mixed cell populations. The expression of immune genes across comparison groups was analyzed using the immune gene list from the ImmPortDB database. Additionally, based on research on interferon responses published by Schoggins et al. (24), the expression of interferon-induced genes in both groups was evaluated through transcriptome sequencing data.

## Statistical analysis

Statistical analyses were performed using R version 4.1.0 and SPSS version 22.0. Violin plots were generated and analyzed using GraphPad Prism version 9.0. For normally distributed continuous data with equal variances, the two-sample *t*-test was applied. Categorical data with small sample sizes or low frequencies were analyzed using Pearson's $\chi^2$ test or Fisher's exact test. Non-parametric median comparisons were conducted using the two-tailed Mann-Whitney *U* test. A *P*-value of <0.05 was considered statistically significant.

## RESULTS

### Demographic and clinical characteristics of SFTS patients

A total of 10 confirmed SFTS patients were enrolled in this study. The cohort included five patients with SAE and five patients without encephalitis. PBMCs were collected for RNA extraction and subsequent transcriptome sequencing analysis. Table 1 outlines the demographic characteristics, underlying conditions, clinical manifestations, and laboratory findings for patients with and without encephalitis. Data are presented as medians with interquartile ranges. No significant difference was observed between the two groups in terms of gender distribution (*P* > 0.05). Although patients in the non-encephalitis group were older than those in the encephalitis group, this age difference did not reach statistical significance (*P* > 0.05).

Hypertension and diabetes were the most prevalent underlying conditions among the SFTS patients, with no significant differences between the two groups. All patients (100%) exhibited high fever and fatigue, with no significant difference in the maximum body temperature recorded between groups. Other common symptoms, such as dizziness, headache, vomiting, diarrhea, and myalgia, showed no statistical significance, which may be attributed to the small sample size of the sequencing group. Laboratory findings at the time of specimen collection are also presented in Table 1. Both groups exhibited a reduction in white blood cell counts, though this difference was not

statistically significant. Notably, the neutrophil percentage was normal in SAE patients but significantly elevated in non-SAE patients ($P$ = 0.008). Additionally, both groups showed a decline in platelet counts, with the encephalitis group having a median of $35 \times 10^9$/L and the non-encephalitis group $50 \times 10^9$/L, but this difference did not reach statistical significance.

## Dynamic laboratory findings in SFTS patients with encephalitis

The natural course of SFTS is characterized by three distinct phases: the febrile stage, the MODS stage, and the convalescence stage (25). The febrile stage (days 0–6) marks the early acute phase of infection, while the second phase (days 7–13) may progress to MODS, a major contributor to disease deterioration and mortality. To elucidate the dynamic profiles of laboratory indicators in patients with SFTS-associated encephalitis, we collected clinical laboratory parameters from both groups during the first two stages and performed statistical analyses. Despite the small sample size, which limited statistical power and led to some non-significant findings, several noteworthy observations emerged.

During the early phase of SFTS (days 0–6), the median serum viral load, alanine aminotransferase (ALT), and aspartate aminotransferase (AST) levels were higher in the encephalitis group compared to the non-encephalitis group, with ALT and AST levels significantly exceeding the normal range (Fig. 1). As the disease progressed to the second stage, viral load decreased markedly in non-encephalitis patients, while encephalitis patients maintained elevated viral copy numbers (Fig. 1), corroborating previous studies (13, 15). Furthermore, compared to non-encephalitis patients, those with SFTS-associated encephalitis exhibited significantly higher lactate dehydrogenase (LDH) levels ($P < 0.01$) and prolonged thrombin time (TT; $P < 0.01$; Fig. 1), indicating the greater severity of the disease.

## Differential gene expression patterns of PBMC in SAE patients compared to non-SAE patients

The CNS complications associated with SFTS are consistently linked to fatal outcomes (25–27). However, the underlying mechanisms remain poorly understood. Given the rapid progression and pronounced variability of MODS in severe fever with thrombocytopenia syndrome, PBMCs were obtained from both SAE and SFTS patients during the MODS phase and subjected to transcriptome sequencing to delineate transcriptomic alterations in SAE patients. Following transcriptome sequencing, differential gene expression analysis was performed to compare the expression profiles of SAE patients to those of non-SAE patients (Table S1). A clustering analysis heatmap encompassing all 244 DEGs was generated (Fig. 2A). Additionally, a volcano plot distinctly illustrated the differential expression of 96 upregulated genes and 148 downregulated genes (|log2 FC| > 1, $P < 0.05$; Fig. 2B). Notably, downregulated genes demonstrated larger fold changes compared to upregulated genes. The results indicated that in the encephalitis group, potassium channel-related gene KCNN3, cell development-related genes EMC8, DANCR, and ASPN, along with the transcription factor ZNF296, were significantly upregulated. Conversely, metabolic-related gene MGAM, immune-related genes LCN2 and IL1R2, the protease inhibitor PZP, and the cytoskeletal protein MYO7A were significantly downregulated.

To further investigate the potential functions of these DEGs, GO and reactome pathway enrichment analyses were performed. The GO enrichment analysis identified the roles and functions of these genes across three categories: biological process (BP), cellular component (CC), and molecular function (MF; Fig. 2C). In the BP category, the DEGs were predominantly associated with extracellular matrix organization, potassium ion transmembrane transport, inflammatory responses, biogenic amine metabolic processes, and innate immune responses. In terms of CC, these genes were mainly localized to the presynaptic membrane, cellular granules, and exosomes. The most enriched MF terms included L-amino acid transmembrane transporter activity, protein

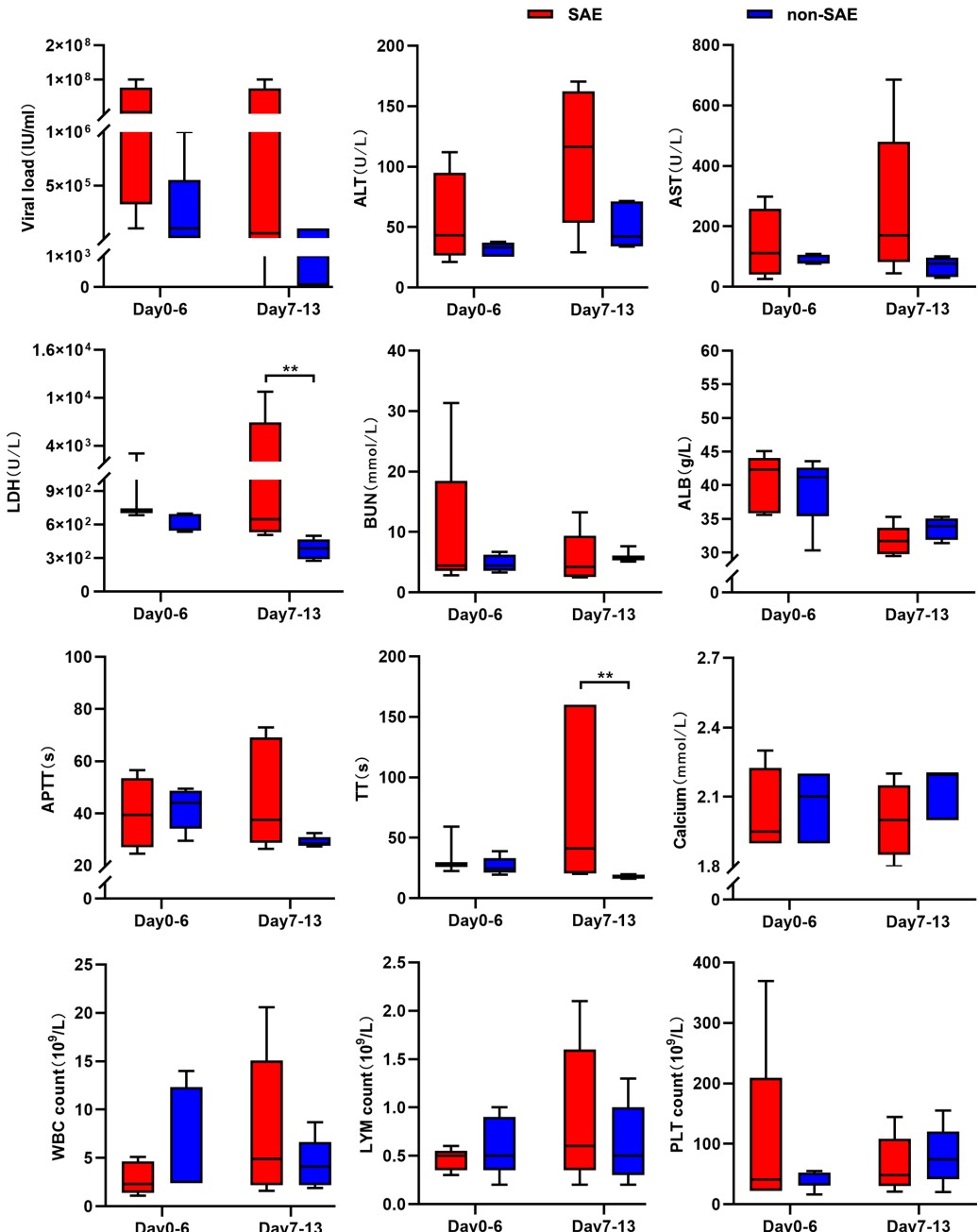

**FIG 1** Dynamic clinical laboratory profiles for SAE patients. Dynamic clinical laboratory findings of serological viral load (normal range, negative), ALT (normal range, 5–40 U/L), AST (normal range, 8–40 U/L), LDH (normal range, 109–245 U/L), blood urea nitrogen (BUN; normal range, 2.9–7.5 mmol/L), albumin (ALB; normal range, 40–55 g/L), activated partial thromboplastin time (APTT; normal range, 25–31.3 s), TT (normal range, 9.8–12.1 s), calcium (normal range, 2.25–2.75 mmol/L), white blood cell (WBC; normal range, 3.5–9.5 × 10⁹/L) count, lymphocyte (LYM; normal range, 1.1–3.2 × 10⁹/L) count, and platelet (PLT; normal range, 125–350 × 10⁹/L) count in SAE and non-SAE patients during the first stage (day 0–6) and the second stage (day 7–13) of disease. All data are presented as median with interquartile range. ** denotes the significance at $P < 0.01$.

tyrosine kinase activity, and calcium ion binding. Furthermore, reactome pathway analysis revealed significant involvement of these genes in neutrophil degranulation, extracellular matrix degradation, collagen degradation, biological oxidation processes, and activation of matrix metalloproteinases (Fig. 2D). Collectively, these DEGs play

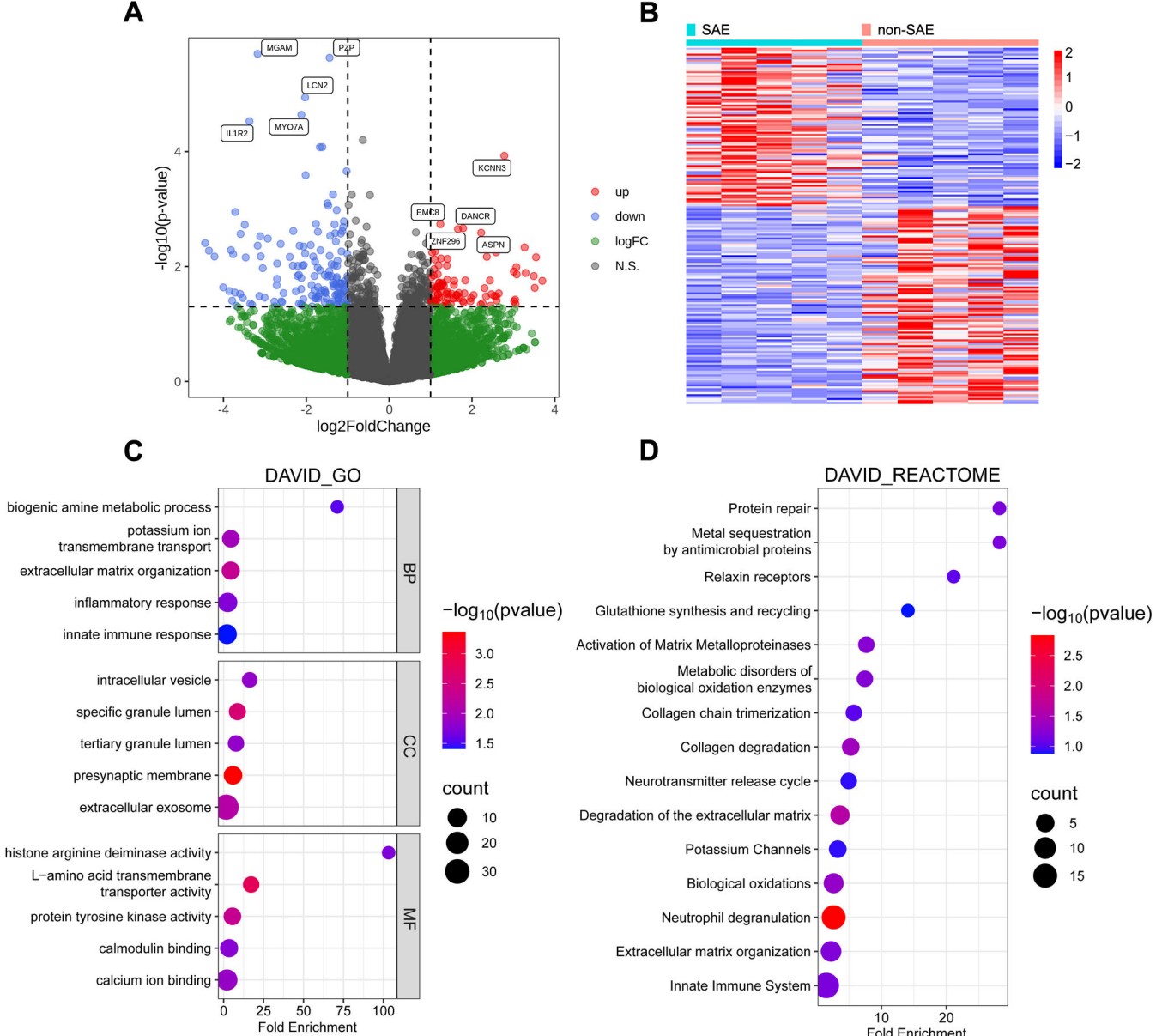

**FIG 2** Identification and functional enrichment analysis of the DEGs in SAE (*n* = 5) and non-SAE (*n* = 5). (A and B) The volcano plot and heatmap display 244 DEGs among the two groups. (C) GO analysis of the DEGs. (D) Reactome pathway enrichment analysis of the DEGs.

vital roles in inflammatory responses, tissue damage, cellular metabolism, and immune function.

## GSEA reveals cellular dysfunction and immune dysregulation in SFTS-associated encephalitis

To provide a more comprehensive analysis of the differential gene expression profiles between patients with SFTS-associated encephalitis and those without, we employed GSEA to identify coordinated gene sets and visualize key biological processes or pathways. Our analysis revealed significant enrichment in the CC terms of GO, hallmark gene sets, and immunological signature gene sets. Notably, gene sets enriched in specific granules, secretory granule membranes, and tertiary granules were downregulated in the non-encephalitis group, consistent with the CC findings from the GO analysis (Fig. 3A). In the hallmark gene sets, genes involved in E2F targets, unfolded protein

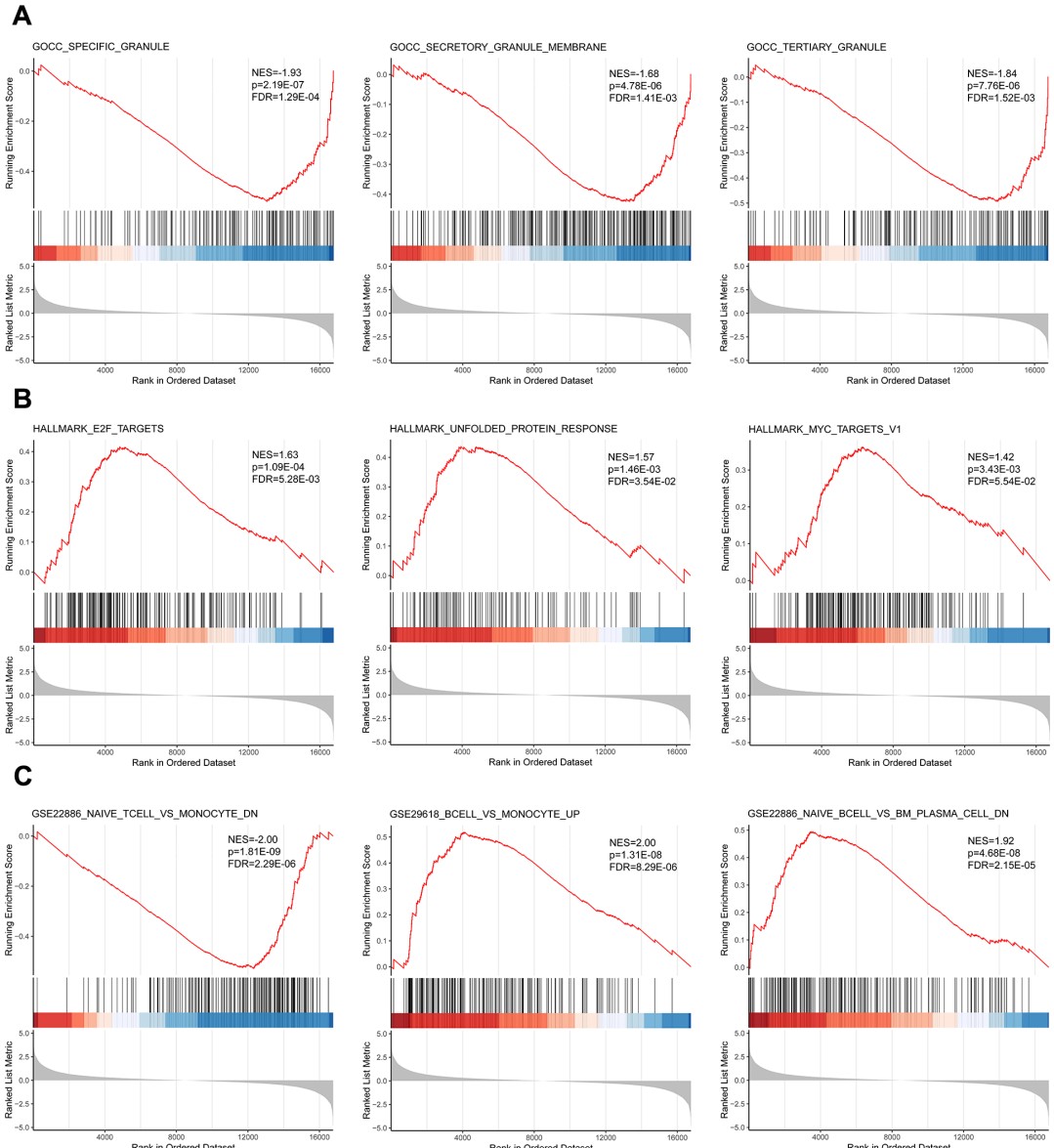

**FIG 3** BP and immunological signature analyses of SAE based on GSEA. (A) Representative pathways enriched in CC gene sets derived from GO terms as determined by GSEA. (B) Representative pathways enriched in hallmark gene sets as determined by GSEA. (C) Representative pathways enriched in the immunological signature gene set as determined by GSEA. Each plot shows the normalized enrichment score (NES), *P*-value, and false discovery rate (FDR) for statistical significance.

response, and Myc targets were significantly upregulated in the encephalitis group, indicating their role in cell proliferation and stress responses (Fig. 3B). Furthermore, the analysis of immunological signature gene sets highlighted significant differences in the immune cell states and perturbations between the encephalitis and non-encephalitis groups (Fig. 3C), suggesting that alterations in host immune responses may play a crucial role in the progression of SFTS-associated encephalitis.

## Immune dysregulation and key immune gene alterations in SFTS-associated encephalitis

The clinical manifestations of SFTS are closely linked to abnormal host immune responses, which are critical in determining disease progression and severity (28–30). Numerous studies have shown that impairments in innate and adaptive immune

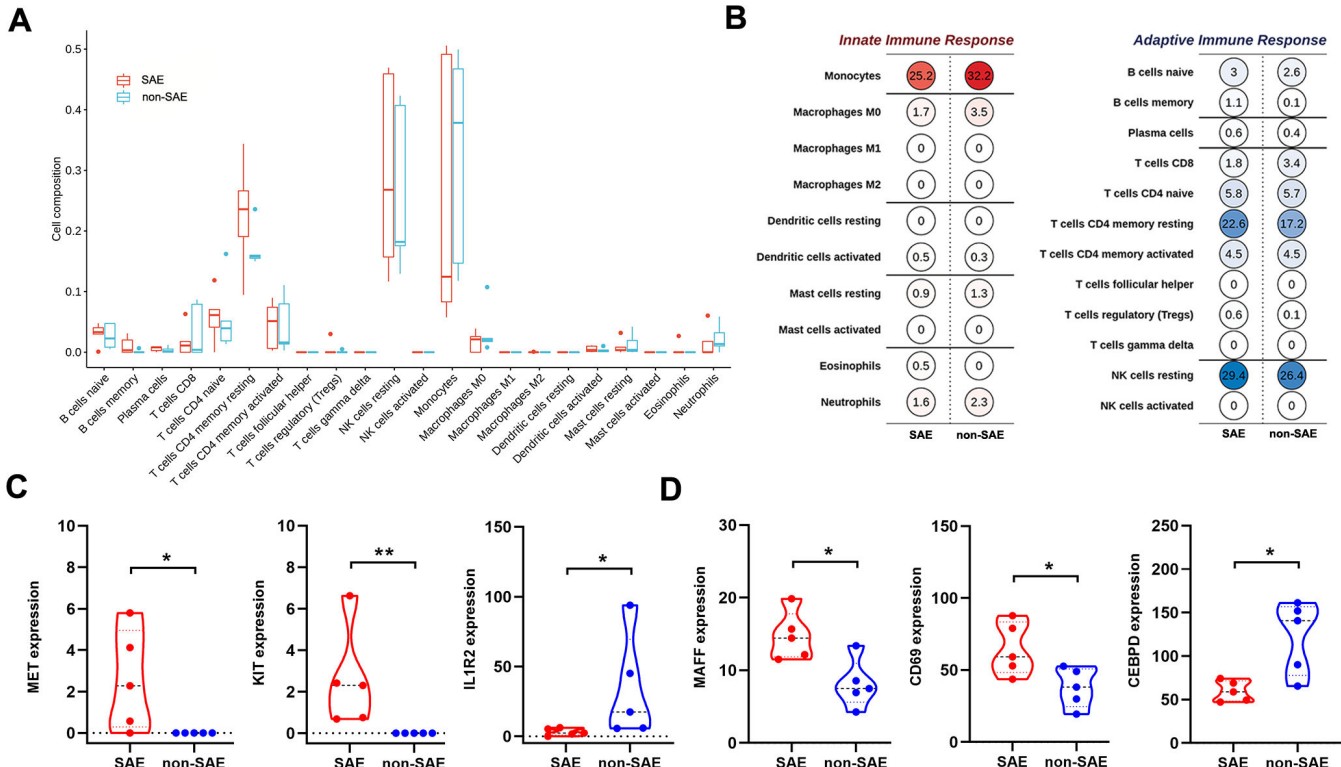

**FIG 4** Immunoinfiltration and immune-related gene analysis. (A and B) Immune cell infiltration in SAE and non-SAE samples. (C) Comparison of significantly different expressed immune-related genes in the SAE and non-SAE groups. (D) Comparison of significantly different expressed interferon-stimulated genes in the SAE and non-SAE groups.

responses are key factors in the fatal progression of SFTS (31, 32). Recent studies have highlighted the significant involvement of monocytes and neutrophils, which are mobilized to infection sites to combat bacterial agents, in the pathogenesis of bacterial meningitis (33). To investigate the immune profile in SAE patients, we employed the Cibersort algorithm to analyze immune cell infiltration characteristics. Our analysis revealed that monocytes, resting natural killer (NK) cells, and CD4+ T memory cells were the most prevalent immune cell types in both groups (Fig. 4A). In contrast, functional CD4+ T cells, CD8+ T cells, and NK cells were markedly reduced (Fig. 4A). Additionally, we categorized these immune cells based on their involvement in innate and adaptive immune response. The immune cell compositions in the SAE and non-SAE groups were similar across both response modes (Fig. 4B). Nevertheless, no significant differences were observed in the relative proportions of the various immune cell types between the two groups.

To further investigate differences in immune gene expression between SAE patients and non-SAE patients, we retrieved an immune gene list from the ImmPortDB database and performed differential gene expression analysis. The results revealed upregulation of the MET proto-oncogene receptor tyrosine kinase (MET) and KIT proto-oncogene receptor tyrosine kinase (KIT) in the encephalitis group, while interleukin 1 receptor type 2 (IL1R2) was downregulated compared to the non-SAE group (Fig. 4C). Previous studies have suggested that SFTSV inhibits the host antiviral immune response by suppressing the activation of the interferon signaling pathway (34). Therefore, we also assessed the expression of interferon-stimulated genes (ISGs) in SAE patients. Our findings indicated that MAF BZIP Transcription Factor F (MAFF) and Early T-Cell Activation Antigen P60 (CD69) were relatively upregulated in the encephalitis group, while CCAAT Enhancer Binding Protein Delta (CEBPD) was downregulated (Fig. 4D). These genes may offer valuable insights into the underlying mechanisms of SFTS-associated encephalitis.

## DISCUSSION

The development of encephalitis significantly contributes to poor prognosis and mortality in SFTS (26). Several studies have confirmed the presence of SFTSV in the cerebrospinal fluid of SAE patients, indicating that the virus can invade the central nervous system and cause intracranial infections and neurological symptoms (12, 35, 36). Currently, SFTS-associated encephalitis is primarily attributed to direct viral invasion and immune-pathological damage induced by cytokines. However, the exact mechanisms and host responses involved remain largely unexplored. In the present study, we enrolled five SAE patients and five non-SAE patients and analyzed their clinical manifestations upon admission and laboratory parameters during the febrile stage and the MODS stage. Owing to the limited sample size, most of the observed differences did not reach statistical significance. Nevertheless, comparative analysis of dynamic laboratory findings revealed multiple abnormal clinical laboratory parameters in SAE patients, particularly elevated LDH levels and prolonged TT. These findings are consistent with previous studies (12, 15). Given the small sample size, non-significant findings should be interpreted with caution and warrant further investigation in larger cohorts.

PBMCs, mainly composed of lymphocytes and monocytes, serve as valuable sources of transcriptomic biomarkers for clinical diagnosis and could reflect global immune response in reaction to external stimuli (37). In this study, we performed whole-genome RNA sequencing of PBMCs derived from patients in the MODS phase to analyze abnormal gene expression in SAE patients compared to non-SAE patients. SFTS patients present marked clinical heterogeneity during the MODS stage. A subset of patients deteriorates rapidly, whereas others remain mildly affected (38). Transcriptomic analysis conducted during this phase is thus essential to delineate gene-expression signatures that distinguish SAE from uncomplicated SFTS. Functional enrichment analysis indicated that the differentially expressed genes were primarily associated with presynaptic membranes, cellular granules, and exosomes and were implicated in pathways related to inflammatory responses, tissue damage, cellular metabolism, and immune regulation. The inflammatory response triggered by SFTSV infection represents a double-edged sword for the host, as excessive inflammation leads to the overproduction of pro-inflammatory cytokines and immune hyperactivation, ultimately contributing to organ dysfunction (39). Previous studies have shown that SFTSV can invade the CNS and may induce immunopathological damage (15). Furthermore, murine models have confirmed that SFTSV can infect A1-reactive astrocytes, replicate in the brain, and subsequently trigger neuroinflammation and brain injury (16, 40). Our findings strongly support these observations and suggest that the development of SFTS-associated encephalitis may be related to the extent of immunopathological damage across different individuals.

To further investigate the differences between encephalitis and non-encephalitis groups, we performed GSEA on the entire transcriptomic profiles. Consistently, we observed significant downregulation of genes associated with cellular granules in non-encephalitis patients, which are involved in pathogen infection, inflammatory regulation, and signal transduction. Analysis of hallmark gene sets revealed significant upregulation of pathways related to cell proliferation and cellular stress in the encephalitis group, potentially associated with CNS injury and MODS induced by SFTSV infection. Moreover, our analysis displayed the immune cell perturbation patterns in the two groups. SFTSV infection disrupted immune homeostasis by suppressing the function of host immune cells while provoking an excessive inflammatory response. This immune dysregulation is characterized by early antiviral immune suppression followed by a cytokine storm, exacerbating pathological damage (41). Interestingly, immune infiltration analysis showed similar immune cell compositions in both groups, with a predominance of resting memory T cells and resting NK cells over their activated counterparts. This phenomenon may reflect immune suppression and evasion mechanisms driven by SFTSV infection.

Considering the immune cell perturbation patterns observed in the GSEA analysis, we speculate that, compared to non-encephalitis patients, encephalitis patients may

not experience substantial alterations in immune cell numbers or composition but rather exhibit changes in immune-related genes. Next, through the analysis of immune genes and ISGs, we identified six key genes: MET, KIT, IL1R2, MAFF, CD69, and CEBPD, which are critically involved in regulating cell proliferation, differentiation, immune modulation, and interferon signaling pathways. Our results indicate that, compared to the non-encephalitis group, the encephalitis group shows significant upregulation of immune-related genes MET and KIT, while IL1R2 is downregulated. The expression of interferon-induced genes MAFF and CD69 is markedly elevated, whereas CEBPD expression is significantly reduced. The MET gene encodes a receptor tyrosine kinase, and its amplification has been linked to reduced STING expression, which compromises interferon responses and inhibits anti-tumor immunity (42). Similarly, KIT, another receptor tyrosine kinase, is essential for cell survival and proliferation (43). IL1R2 encodes interleukin-1 receptor 2, functioning as an antagonist to IL-1 receptors, thereby inhibiting IL-1-induced inflammatory responses (44, 45). The decreased expression of IL1R2 in the encephalitis group may remove the negative regulation on the IL-1 signaling pathway and contribute to excessive inflammation. MAFF, a transcription factor, correlates with inflammatory responses and immune cell infiltration in certain cancers (46). CD69 is expressed in various immune cells. Evidence indicates that CD69 enhances the activation and cytokine secretion of T cells, B cells, and NK cells (47, 48). Furthermore, CD69 possibly promotes cerebral thrombus formation in mice via regulating von Willebrand factor (49). CEBPD encodes a transcription factor involved in differentiation and inflammation. It has been shown to induce the expression of secretory factors in astrocytes and affect neuronal apoptosis and inflammation (50). These gene expression profiles provide new molecular insights into the mechanisms of encephalitis pathogenesis. Aberrant activation of receptor tyrosine kinase signaling may drive abnormal proliferation of neuroimmune cells, while imbalances in interferon responses and dysregulation of inflammatory control may collectively disrupt immune homeostasis in the central nervous system. The dysregulation of these genes may serve as potential contributors to CNS injury induced by SFTSV infection. Further investigations are required to elucidate the functional roles of these genes in the pathogenesis of SFTS-associated encephalitis.

Several limitations exist in this study. Due to the challenges in acquiring clinical samples from SFTS patients with encephalitis, the sample size for transcriptomic sequencing was relatively limited. This limitation may affect the statistical power of differential gene expression screening and functional enrichment analysis, underscoring the need for an expanded cohort to validate these findings. A matched healthy control cohort is absent. Public data cross-validation was infeasible because baseline characteristics of existing SFTS data sets diverge, and no SAE data are available. Future studies will rectify this limitation by expanding the sample size, enrolling uninfected controls, and employing quantitative real-time PCR for independent verification. Moreover, this study primarily relied on bioinformatics predictions and statistical analyses, lacking *in vitro* or *in vivo* experimental validation of key gene mechanisms. This gap restricts a deeper understanding of the pathogenic mechanisms involved. Furthermore, the sequencing data were mainly derived from single-time point sample collections, which restricts the dynamic assessment of immune responses and disease progression. Longitudinal cohort studies are necessary to better understand the interaction between host immune responses and disease severity throughout disease progression.

Taken together, our findings provide potential research targets for understanding the pathogenesis of SFTS-associated encephalitis, which highlight the potential role of immune microenvironment dysregulation in SFTS-induced neural injury. Our study will contribute to the broader understanding of SFTSV pathogenesis and may pave the way for the development of targeted interventions to improve outcomes in SFTS patients with CNS involvement.

## ACKNOWLEDGMENTS

This work was supported by the National Key Research and Development Program of China (2023YFC2309100), State Key Laboratory for Diagnosis and Treatment of Severe Zoonotic Infectious Diseases (2024KF10013), project of Institute of Chinese medicine, Nanjing University (ICM2024019), and Science Fund for Distinguished Young Scholars of Jiangsu Province (BK20250016).

## AUTHOR AFFILIATIONS

[1]Department of Clinical Laboratory, Nanjing Drum Tower Hospital, Clinical College of Jiangsu University, Nanjing, Jiangsu, China

[2]School of Environmental and Biological Engineering, Nanjing University of Science and Technology, Nanjing, Jiangsu, China

[3]Department of Gynecology and Obstetrics, Nanjing Drum Tower Hospital, The Affiliated Hospital of Medical School, Nanjing University, Nanjing, Jiangsu, China

[4]Center for Infectious Diseases, Vision Medicals Co., Ltd, Guangzhou, Guangdong, China

[5]Department of Laboratory Medicine, Joint Institute of Nanjing Drum Tower Hospital for Life and Health, College of Life Science, Nanjing Normal University, Nanjing, Jiangsu, China

[6]State Key Laboratory for Diagnosis and Treatment of Severe Zoonotic Infectious Diseases, Wuhan, Hubei, China

## AUTHOR ORCIDs

DaiQing Wu http://orcid.org/0009-0004-3403-8258
Yuxin Chen http://orcid.org/0000-0001-5955-687X

## AUTHOR CONTRIBUTIONS

DaiQing Wu, Formal analysis, Methodology, Writing – original draft, Writing – review and editing | AoFan Wang, Formal analysis, Writing – original draft, Writing – review and editing | Junjie Shi, Data curation, Software, Visualization | Ying Zhang, Writing – original draft | Yu Geng, Formal analysis, Methodology | Huifang Liu, Formal analysis, Methodology | Yuanyuan Wu, Investigation, Resources | Wenwen Kong, Investigation, Resources | Yijia Zhu, Supervision, Validation, Visualization | Yuxin Chen, Conceptualization, Funding acquisition, Project administration, Resources, Supervision, Validation, Visualization, Writing – review and editing

## DATA AVAILABILITY

The raw sequence data reported in this paper have been deposited in the Genome Sequence Archive in National Genomics Data Center, China National Center for Bioinformation/Beijing Institute of Genomics, Chinese Academy of Sciences (GSA-Human: HRA013026), and are publicly accessible at https://ngdc.cncb.ac.cn/gsa-human (51, 52). Other contributions presented in the study are included in the table and figures; further inquiries can be directed to the corresponding author.

## ETHICS APPROVAL

Ethical approval for the study was granted by the Human Ethics Committee of Nanjing Drum Tower Hospital (2022-238-02). Due to the retrospective nature of the study, informed consent was waived.

## ADDITIONAL FILES

The following material is available online.

## Supplemental Material

**Text S1 (Spectrum01161-25-s0001.txt).** Sample identifiers and their corresponding clinical group classification (SAE or non-SAE), used for differential gene expression and downstream bioinformatics analyses.

**Text S2 (Spectrum01161-25-s0002.txt).** Normalized counts of all detected genes across 10 patient samples (SAE and non-SAE), generated by DESeq2.

**Table S1 (Spectrum01161-25-s0003.xlsx).** Differentially expressed genes in patients with SFTS-associated encephalitis.

## Open Peer Review

**PEER REVIEW HISTORY (review-history.pdf).** An accounting of the reviewer comments and feedback.

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
