## [Reviewer comments · Microbiology Spectrum]

Microbiology Spectrum

Blood transcriptomic profiling reveals gene expression alterations in patients with SFTS-associated encephalitis

Dai-Qing Wu, Aofan Wang, Junjie Shi, Ying Zhang, Geng Yu, Huifang Liu, YuanYuan Wu, WenWen Kong, Yijia Zhu, and Yuxin Chen

Corresponding Author(s): Yuxin Chen, Department of Laboratory Medicine, Nanjing Drum Tower Hospital, The Affiliated Hospital of Medical School

Review Timeline:

Submission Date:	April 14, 2025
Editorial Decision:	June 30, 2025
Revision Received:	August 18, 2025
Accepted:	August 21, 2025

Editor: Peter Pelka

Reviewer(s): The reviewers have opted to remain anonymous.

Transaction Report:

DOI: <https://doi.org/10.1128/spectrum.01161-25>

Re: Spectrum01161-25 (**Blood transcriptomic profiling reveals gene expression alterations in patients with SFTS-associated encephalitis**)

Dear Prof. Yu-xin Chen:

Thank you for the privilege of reviewing your work. Below you will find my comments, instructions from the Spectrum editorial office, and the reviewer comments.

I have now received the reviews for your manuscript. The reviewers found the manuscript of interest but highlighted a few limitations that will need to be addressed. Particularly important is submission of all transcriptomic data into a public database which must be done prior to resubmission. Providing a full list of differentially regulated genes as a supplemental spreadsheet is also essential.

Revision Guidelines

Sincerely,
Peter Pelka
Editor
Microbiology Spectrum

Reviewer #1 (Comments for the Author):

Major Points:

1. The author mentioned that the course of SFTS is divided into three typical stages: the febrile stage, multiple organ dysfunction syndrome (MODS) stage, and convalescence stage. During different stages, the transcriptional profiles of peripheral blood cells may undergo dynamic changes. Please specify in the manuscript the stage from which the currently analyzed samples were derived, and discuss whether this could impact the conclusion or what implications it may have for diagnosis or prognosis.
2. This study included a limited number of samples and lacked uninfected or normal controls. The identified differential transcriptomic changes, especially key differential genes, have not been validated. It is recommended to analyze published SFTS patient transcriptome data (such as in PMID: 34561445, 33593977, 34818556, 40263930) to validate the major findings.

Reviewer #2 (Comments for the Author):

Using RNA sequencing of PBMCs, the authors analyzed gene expression profiles in SAE patients compared to non-SAE patients with SFTS. Overall, the manuscript is written and presented well.

1. please provide the lists of up- and down-regulated genes in a supplemental file.
2. line 222, please provide the normal range either in the figure legend or in the article. Not every reader is familiar with the parameters.
3. line 225, please provide reference after "previous studies".
4. please spell out all the abbreviations in the figure 1 legend. Again, not every reader is familiar with all the parameters. It's easier to read without going back and forth between the figure and main article.

Response to Reviewers

Thank you for your insightful comments and constructive suggestions. Below, we address each reviewer's comment in sequence. Reviewer's comments are in black, and our responses are in red.

Reviewer #1

Comment 1: The author mentioned that the course of SFTS is divided into three typical stages: the febrile stage, multiple organ dysfunction syndrome (MODS) stage, and convalescence stage. During different stages, the transcriptional profiles of peripheral blood cells may undergo dynamic changes. Please specify in the manuscript the stage from which the currently analyzed samples were derived, and discuss whether this could impact the conclusion or what implications it may have for diagnosis or prognosis.

Response 1: Thanks for your comment. The PBMC specimens used for sequencing were obtained from SFTS or SAE patients during the MODS phase, and this information has been added to the main text. These points have been appended to the Discussion: SFTS patients presents marked clinical heterogeneity during MODS stage. A subset of patients deteriorate rapidly, whereas others remain mildly affected. Transcriptomic analysis conducted during this phase is tessential to delineate gene-expression signatures that distinguish SAE from uncomplicated SFTS.

Comment 2: This study included a limited number of samples and lacked uninfected or normal controls. The identified differential transcriptomic changes, especially key differential genes, have not been validated. It is recommended to analyze published SFTS patient transcriptome data (such as in PMID: 34561445, 33593977, 34818556, 40263930) to validate the major findings.

Response 2: We sincerely appreciate your highlighting the limited sample size and absence of uninfected or healthy controls. We fully agree with this limitation and have now explicitly addressed it in the Discussion. We carefully assessed the feasibility of validating our findings with publicly available datasets. However, the following constraints precluded such an approach: published SFTS cohorts exhibit substantially

different baseline characteristics from our study population, and matched transcriptomic data from SAE patients are unavailable. Moreover, public datasets were generated using alternative library-preparation platforms (Illumina vs. BGISEQ), RNA-extraction protocols, and bioinformatic pipelines, introducing substantial technical heterogeneity. In the absence of adequate controls, batch-effect correction would be unreliable and could yield misleading conclusions.

In light of these constraints, we have added the following paragraph to the Discussion: And a matched healthy control cohort is absent. Public data cross-validation was infeasible because baseline characteristics of existing SFTS datasets diverge and no SAE data are available. Future studies will rectify this limitation by expanding the sample size, enrolling uninfected controls, and employing qPCR for independent verification.

Reviewer #2

Comment 1: please provide the lists of up- and down-regulated genes in a supplemental file.

Response 1: Thanks for your comment. Up- and down-regulated gene lists are provided in Supplementary sheet 1.

Comment 2: line 222, please provide the normal range either in the figure legend or in the article. Not every reader is familiar with the parameters.

Response 2: Thanks for your comment. Normal ranges for clinical laboratory parameters are provided in the corresponding figure legends (Figure 1).

Comment 3: line 225, please provide reference after "previous studies".

Response 3: Thanks for your comment. The missing references have now been inserted at the appropriate locations.

Comment 4: please spell out all the abbreviations in the figure 1 legend. Again, not every reader is familiar with all the parameters. It's easier to read without going back and forth between the figure and main article.

Response 4: Thank you for pointing this out. All abbreviations are now fully defined in the legend to Figure 1.

Re: Spectrum01161-25R1 (**Blood transcriptomic profiling reveals gene expression alterations in patients with SFTS-associated encephalitis**)

Dear Prof. Yuxin Chen:

Your manuscript has been accepted, and I am forwarding it to the ASM production staff for publication. Your paper will first be checked to make sure all elements meet the technical requirements. ASM staff will contact you if anything needs to be revised before copyediting and production can begin. Otherwise, you will be notified when your proofs are ready to be viewed.

Sincerely,
Peter Pelka
Editor
Microbiology Spectrum